# OpenReview forum: "MONA: Myopic Optimization with Non-myopic Approval Can Mitigate Multi-step Reward Hacking"
_ICML.cc/2025/Conference — ICML 2025 poster_

### Official Review · Reviewer_cufo · 2025-03-14

**Overall Recommendation:** 4

**Summary:**

The paper investigates the problem of multi-step reward hacking, in which an agent trained with an ordinary reinforcement learning algorithm, due to the statement of the RL optimization problem, learns to optimize for the sum of current and future rewards, potentially learning multi-step strategies to increase its long-horizon reward in ways that are very hard for overseers to detect, compared to e.g. single-step reward hacking. In particular, the paper introduces a way of doing Myopic Optimization with Non-myopic Approval (MONA), in which the optimization objective, rather than consisting of just the instantaneous rewards, has an additional non-myopic approval reward, which represents the overseer's estimation on how much taking the action improves the agent's future prospects. Notably, the non-myopic approval cannot depend on actual observations of the action's consequences, as that would make this no different from the normal RL optimisation objective, in the sense of the agent having an incentive to control *future* states and rewards, which the MONA agent does not. The paper then presents empirical evidence in the form of three experiments—test-driven development, loan application evaluation, and a block-moving task—in which ordinary reinforcement learning leads to interesting multi-step reward hacking in the form of writing easy tests for future self, basic steganography, and confusing the reward-assigning overseer, respectively—and demonstrate that identical agents trained with a MONA objective, while (as expected) obtaining significantly less observed reward during training, perform much better in the ground-truth cases, and, in some of the cases, demonstrate that the MONA-trained agents do not reward hack.

**Claims And Evidence:**

Yes, the claims made in the paper are generally very well-supported by theoretical and empirical evidence. There are no problematic claims that I noticed.

**Essential References Not Discussed:**

References are good

**Experimental Designs Or Analyses:**

I think the experimental designs are valid and well-done. I especially like the analysis shown in figure 4(c), where the counterfactual impact of the nationality is directly analyzed.

**Methods And Evaluation Criteria:**

Yes; all three experiments are in environments where reward hacking is a natural outcome (that is, in all of the environments, the rewards are "mis-specified" in such a way that a non-myopic agent would figure out an undesired way to increase reward.)

**Other Comments Or Suggestions:**

I found myself wishing that the Camera Dropbox task got a better explanation—in section 4.1, the description didn't really make sense to me, even after looking at Figure 10 (why is pushing multiple boxes into the hole bad? How can you block the camera and push the very same box into a hole? what are "multiple rewards"?) This finally made sense to me after I read the first 4 sentences in D3; I would just move those to the main paper.

It might also be nice to have a more explicit argument for why single-step reward hacking is "better"—this isn't super intuitively obvious to me, especially when you mention in Section 3.3 that non-myopic approval can be implemented by modelling a reward function that's based on human preferences/scores/feedback—aren't RMs kind of notoriously reward-hackable in difficult-to-detect ways?

**Other Strengths And Weaknesses:**

I think that overall the paper presents a very strong approach for dealing with multi-step reward hacking, and deals very well with the difficulties of formalization of what this means, while presenting strong experimental evidence to support the method.

Section 4.4 is very interesting—in fact, this is the main part of the paper that I'm interested in: the reason being that if I had some sort of RL task for which I was suspicious of multi-step reward hacking being possible and I wanted to use MONA, it is sort of unclear to me what kind of reward I should use such that the model still performs fine in the potentially long-horizon ground truth setting but does not do multi-step reward hacking. Essentially, in section 2, the paper claims to "attempt to start a discussion of safety-performance tradeoffs: knowingly accept a possible reduction in observed performance to improve potentially unobservable properties". I think the existence of section 4.4 is a great strength of the paper, but it's a significant weakness, in my opinion, that the actual safety-performance tradeoff is underdiscussed. There is likely a meaningful set of Pareto frontiers here between how much implicit consideration of future outcomes the MONA reward function has and how much this affects performance at different thresholds for the acceptability of multi-step reward hacking, and this is the sort of thing that would make me significantly more excited about the applicability of this method outside of toy problems.

Another thing which I would like the main paper to have discussed more is section C3; while the paper does not explicitly claim that MONA has any sort of guarantees on getting rid of reward hacking, Theorem 3.1 feels sort of misleading without the corresponding discussion; in fact, I spent a fair amount of time thinking about what the theorem actually implies, thinking of how I would write in my review that this doesn't seem to guarantee anything about multi-step reward hacking not happening, and then seeing that this is already addressed in the supplemental material (I realize there were significant space constraints, but hopefully you can fit in a discussion of this for the camera-ready version!)

**Questions For Authors:**

I don't have any questions where the response would be likely to significantly change my evaluation of the paper

**Relation To Broader Scientific Literature:**

Reward hacking has been observed in many contexts, leading to potentially scary multi-step behaviors like steganography (e.g. Motwani et al, 2024). This paper presents a way to decrease the likelihood of multi-step reward hacking occurring, which significantly contributes to reinforcement learning generally, agent evaluations, and AI alignment.

**Theoretical Claims:**

The proof of theorem 3.1 seems straightforwardly correct, if not particularly strong in practice.

---

> ### Author Rebuttal · Authors · 2025-04-01
>
> We are happy you found the paper insightful, and thanks for the detailed comments on improving the clarity in the main paper.
>
> We agree that the most important open question about MONA is what the safety-performance Pareto frontier looks like in practice and we’re hoping this will be studied in future work!
>
> Unfortunately, we weren’t able to discuss all of the interesting aspects of MONA in the main paper. But we agree that Appendix C3 contains an important result and we will highlight this more in the main paper. We’ll also improve the description of the Camera Dropbox environment. Thanks for the concrete suggestion for doing this!

---

### Official Review · Reviewer_6k8K · 2025-03-14

**Overall Recommendation:** 3

**Summary:**

The paper introduces a method, called MONA, that mitigates multi-step reward hacking by limiting optimization to be myopic and adding a hand-crafted non-myopic approval reward. The paper provides three case studies to demonstrate how MONA can avoid multi-step reward hacking in comparison to ordinary RL and provides some insight on how even inaccurate foresight can work.

**Claims And Evidence:**

The paper motivates their method by claiming that it "restricts the the agent to strategies within the spotlight [of strategies that human experts can understand]" on line 025. It's not clear that restricting to myopic optimization will actually produce such strategies. None of the experiments provide evidence to this fact.

The paper claims that MONA has only a "reasonable performance cost" and can still lead to superhuman agents. The experiments provided are not convincing in arguing that myopic optimization will be sufficient for this.

The experiments do a good job of supporting the claims that MONA prevents multi-step reward hacking. However, the evaluation is hurt by relying solely on benchmarks where it's clear that the environment was designed so that multi-step optimization will lead to reward hacking. It would be nice to see MONA more broadly tested in existing reward hacking benchmarks to understand its limitations.

**Essential References Not Discussed:**

None.

**Experimental Designs Or Analyses:**

In the "Test-driven Development" experiment, the LM is prompted to be a "bad programmer" and write an "incorrect solution". This is a strange way to prompt an LM for programming tasks and doesn't match how a LM would be used or finetuned for such a task. It's not clear to me how this might confound the results. I would feel more comfortable about this experiment if instead a weaker LM were used that couldn't solve all of the tasks to begin with even when unprompted.

The setup for the Camera experiment seems a bit backwards and is not a good reflection of a real-world scenario. Here, the non-myopic approval required by MONA is dependent on having the value function of a non-reward hacking policy, in which case we've already solved the problem. I also don't think the application of noise in this scenario is compelling: isn't the optimal (myopic) policy the same regardless of how much noise is applied to the value function?

**Methods And Evaluation Criteria:**

The benchmarks are well designed for highlighting the potential problems of multi-step reward hacking. However, the evaluation seems overly designed to situations in which myopic optimization will succeed and the reward has been specifically setup in a way to elicit reward hacking. A broader evaluation is necessary to understand the limitations of MONA and myopic optimization.

**Other Comments Or Suggestions:**

The definition of reward hacking given at the start of section 2 is confusing and unsatisfying. What does "understood" refer to? What does the system designer need to understand? Since reward hacking is such a central aspect of the paper, I would like to see the paper engage with other definitions of reward hacking in the literature and carefully define how their method relates to these definitions.

**Other Strengths And Weaknesses:**

Weaknesses:

The paper does a poor job of engaging with how much myopic optimization might limit the capabilities of the model. I think a more thorough investigation of this to understand the limits of MONA would substantially strengthen the paper.

As discussed in other parts of my review, I think the evaluations are overly hand-designed and don't provide convincing evidence that this approach will work in general.

Strengths:

The paper is well written and easy to follow. I think the chosen evaluations do a good job of highlighting setting in which myopic optimization can help prevent reward hacking.

**Questions For Authors:**

1. It's difficult to parse the performance of MONA in Fig 7. In the top plot, is the value of MONA given by the first x-value (mainly failing) or the second x-value (full desired behavior). If it's the latter, what does x=0 represent (i.e. an optimization horizon of zero)?

2. It feels like the formulation of MONA is simply pushing the burden of reward hacking onto the non-myopic approval. The non-myopic approval still needs to be carefully designed to prevent its own form of reward hacking. Indeed, if the non-myopic approval somehow related to future states in the trajectory and is not perfectly specified, it could also lead to multi-step reward hacking (as mentioned in Section 4.4). Why does delegating the non-myopic aspect of the reward function to this auxiliary reward fundamentally mitigate the reward hacking problem?

2. The paper repeatedly states that it's desirable to have policies that humans can understand so that they can have proper oversight. I understand that myopic optimization will make policies weaker, but why will it necessarily make the policies easier to understand?

**Relation To Broader Scientific Literature:**

The paper relates closely to other existing ideas of using human-designed approval to mitigate reward hacking (e.g. decoupled approval) but extends these ideas by including a myopic version of the underlying reward.

**Theoretical Claims:**

I checked theorem 3.1 and it is sound. The result follows immediately from the construction of the CID.

---

> ### Author Rebuttal · Authors · 2025-04-01
>
> Thanks for your review and the constructive criticism!
>
> We broadly agree with the key limitations that your review identifies (and we discuss them in Appendix B). It will be important for future work to study the competitiveness of MONA and to test MONA in more realistic environments.
>
> However, these questions were out-of-scope for the present paper, which already provides multiple major contributions. Our work is the first work to demonstrate multi-turn reward hacking during RL with LLMs; the first work to show steganography arising during end-to-end RL; and, the first work testing the effect of myopic optimization to address it. We believe these contributions are novel, impactful, and of high interest to the ICML community.
>
> Our environments are designed to make it easy to study multi-turn reward hacking. The main reason for this is that multi-turn reward hacking can be difficult to detect in practice and the practical use of multi-turn RL with LLMs is very limited at the moment. As multi-turn RL is used in more realistic environments, it will become feasible to study the competitiveness of MONA in practical settings.
>
>
> Let us also respond to specific questions you raised below.
>
> > I would feel more comfortable about this experiment if instead a weaker LM were used that couldn't solve all of the tasks to begin with even when unprompted.
>
> We use this unusual prompt because current LLMs have been heavily finetuned for coding which leaves little room for improvement via RL training. We tried using a smaller LM for this task, but could not find a model that cannot solve the tasks but can learn to do so via RL training (likely because coding performance in these models is already strongly elicited via finetuning).
>
> > The setup for the Camera experiment seems a bit backwards and is not a good reflection of a real-world scenario. [...] I also don't think the application of noise in this scenario is compelling: isn't the optimal (myopic) policy the same regardless of how much noise is applied to the value function?
>
> Using a non-reward hacking policy does, of course, give MONA an unfair advantage. The point of this experiment is not to check if MONA performs better than RL without this value function, but to study how well MONA works with noisy approval feedback.
>
> Applying noise to the value function _does_ change the optimal (myopic) policy. While the value function is the expected future reward of the non-reward hacking policy, we apply noise after the computation of the expectation. This can change the optimal policy if the relative order of states’ values changes.
>
>
> > is the value of MONA given by the first x-value (mainly failing) or the second x-value (full desired behavior). If it's the latter, what does x=0 represent (i.e. an optimization horizon of zero)?
>
> The second x value (x=1) corresponds to MONA. In our implementation, x=0 optimizes an all-zero reward function, so it effectively gives a random policy. This is indeed confusing, we’ll make sure to clarify this when revising the paper.
>
>
> > Why does delegating the non-myopic aspect of the reward function to this auxiliary reward fundamentally mitigate the reward hacking problem?
>
> The approval reward is not the primary reason MONA helps with multi-turn reward hacking. In most of our experiments (except the gridworld), we use the same reward function with MONA and ORL (also cf. our response to reviewer xDwo). The MONA agent does not get additional information compared to the ORL agent. Instead, MONA helps prevent reward hacking because of the way the rewards are propagated across the trajectory.
>
> MONA does not address single-turn reward hacking, i.e., the agent might still try to achieve high approval feedback in unintended ways. We believe reducing reward hacking to single-step reward hacking is progress because we expect single-step reward hacking to be significantly easier to detect (as discussed in Appendix B).
>
> We do not claim that MONA “fundamentally mitigates the reward hacking problem”. We claim that it removes the incentive for multi-step reward hacking which is particularly difficult to detect.
>
>
> > The paper repeatedly states that it's desirable to have policies that humans can understand so that they can have proper oversight. I understand that myopic optimization will make policies weaker, but why will it necessarily make the policies easier to understand?
>
> We typically think of the approval feedback as provided by human overseers. Approval means that humans recognise the value of this action for solving the task at hand. We expect this will make the agent’s actions more understandable to humans because the overseer will likely not approve actions that they don’t understand.
>
> Of course, this effect will depend on how exactly the approval feedback is constructed. We discuss this to some extent in Appendix B where we compare different approaches to constructing approval feedback. But, we will make sure to clarify this point in the main paper.

---

### Official Review · Reviewer_xDwo · 2025-03-17

**Overall Recommendation:** 4

**Summary:**

This paper introduces an approach (MONA) to mitigating the risk of multi-step reward hacking (where an agent executes undesirable, multi-step plans to achieve high reward) by decomposing the reward function into a myopic task reward and an non-myopic approval reward. They demonstrate that this can help reduce multi-step reward hacking in two short-horizon text-based tasks using LLM agents and one longer-horizon grid-world task using dynamic programming/RL.

### Update after rebuttal

I am not changing my score. The authors have replied to my questions and minor concerns adequately, and I still think the paper should be accepted, though I am not quite prepared to move from "Accept" to "Strong Accept".

**Claims And Evidence:**

While multi-step reward hacking is not yet prevalent, the authors did a good job of motivating the importance of this problem without over-claiming. Their proposed approach is supported both by some straightforward (but still helpful) theoretical results and a range of empirical results with numerous variations and ablations.

One claim that I did not fully understand was:

> This is a non-trivial modification, and we believe it affects exploration and learning in subtle ways because that part of policy-space is less heavily modified by supervized fine-tuning and reinforcement learning before we begin our own training.

in reference to "The model ... being told it is a “bad programmer” that should write an “incorrect solution” to cause it to perform poorly". What is the "part of policy-space" that the authors are referring to? Is it that which is conditioned on this prompt? While I agree that SFT and RL may not target exactly situations in which an agent has been told this, I would have thought that, e.g., there is still quite a lot of overlap or transfer (e.g., because the model is the context of having been presented a programming problem to solve).

**Essential References Not Discussed:**

In general, I think the authors did a good job of discussing related work. As someone reasonably familiar with this area, I didn't spot any glaring omissions. It perhaps would have been nice to see references to some of the earlier, less formal discussions on this topic by Hubinger, Demski, etc., but I'm not sure I'd classify those as _essential_.

**Experimental Designs Or Analyses:**

I did check the experimental designs and analyses and they seem sound to me.

I have already mentioned that one of the main issues is that the LLM tasks only have two steps.

I also noted that although the authors run RL experiments for the gridworld task their results in the main body use value iteration. I assume this is probably because for the different variations and ablations value iteration is less noisy than PPO, but it would be good if the authors could clarify the reasoning behind their decision here.

**Methods And Evaluation Criteria:**

The chosen evaluation methods and baselines do make sense, although it would have been great to see slightly longer-horizon tasks for the LLM agents. This would have been closer to the problem setting the authors care about and would have also (I imagine) been more interesting when it came to the specification of the approval vs. myopic rewards, which is – to the best of my understanding – one of the main challenges of actually implementing MONA.

My guess is that the authors don't do this because it is hard to get RL working for LLMs on longer horizon tasks (as they correctly note, theirs is the first time that RL-based steganography in LLMs has been observed, despite numerous past attempts). Still, with it now being possible to create reasoning models using RL, I would be interested to see how their methods apply there.

**Other Comments Or Suggestions:**

Typos:

- I don't think there should be a dotted information edge from $s_i$ to $a_{i+1}$ in Figure 2.b). (As a side-note I don't think you actually explain the CID notation anywhere, not that it is really essential to understanding the diagram.)
- In D.2.2: "State and transition functions The initial"
- It's "parsable", not "parse-able"

**Other Strengths And Weaknesses:**

Overall I thought this was a good paper. The idea behind MONA is a natural one but well-executed. The paper is clear and well-presented, and the results justify the authors' approach. While in some ways this is still preliminary work and there are lots of open questions, I think it's important that other researchers have something to build on, and that the challenges with myopic vs. non-myopic supervision are more widely appreciated. I also appreciated some of the more minor results in the paper, such as the demonstrations of RL-based steganography in LLMs and MONA unlearning reward hacking behaviour in the grid-world experiment.

Aside from the issue about it ideally being nicer to have longer-horizon LLM experiments, there are two other issues I had with the paper, one of which is concrete and one of which is admittedly quite vague, and more of an open question to the authors:

1. I found it kind of confusing that the paper is premised around the idea that an agent's reward can helpfully be decomposed into $r^I$ and $r^A$ but then in the LLM experiments there is no such decomposition. Rather, $r^A$ seems to be implicitly captured by some post-hoc ground-truth evaluation that highlights the shortcomings of the reward hacking policy. The authors note that "in practice, we may choose to implement a mechanism to provide $r_{\text{MONA}}$ without explicitly distinguishing between $r^I$ and $r^A$" but at least for the purpose of exposition I would suggest more clearly writing down how the reader should interpret these two reward structures for the experiments in the paper (even if only in a footnote).
2. It seems that the efficacy of MONA fundamentally depends on how the approval reward is specified, as this is what makes MONA agents more competitive compared to standard agents that receive non-myopic rewards. Throughout the paper I had an uneasy intuition that we were sort of just 'passing the buck' of reward specification to the approval reward, and moreover that having an approval reward capturing what the overseer views to be "reasonable" isn't all that robust. E.g. for the loan applicant summaries in Figure 6, both of these seem pretty reasonable to me, at first glance. I'd be interested to hear the authors' response to this point, as I might be missing something.

**Questions For Authors:**

1. Please could you clarify the claim that "This is a non-trivial modification, and we believe it affects exploration and learning in subtle ways because that part of policy-space is less heavily modified by supervized fine-tuning and reinforcement learning before we begin our own training." as discussed further above?
2. My (admittedly limited) understanding is that current approaches to produce reasoning LLMs via RL work simply by using a sparse end-of-task reward rather than by direct process supervision. To the extent that step-by-step rewards are hard to create and/or don't actually work as well, how well do the authors expect their approach will transfer to these new methods of training more powerful foundation model agents that can plan over  longer time horizons?
3. In Figure 4.a), if "MONA achieves roughly the best score possible without considering nationality" and the labels are only swapped on 30% of data points, why is MONA's return approximately 0.7/2 instead of 0.7 (compared to ORL's return of 1)?
4. In appendix D.1.5, whats is the "TFO" agent? I couldn't find the definition of this acronym anywhere.
5. I didn't understand the explanation of the results in appendix F.2. Shouldn't the ORL agent end up with a higher coordination advantage compared to MONA?
6. This is more of a high-level question, but do the authors see their method as mainly combatting reward-hacking in training or when during deployment, or both? To what extent is there a meaningful distinction here, and which do the authors think is the most important issue? For example, I can imagine an agent trained to be very competent at pursuing long-term plans as being more dangerous, even if given myopic objectives once deployed, but also that a myopically trained agent might eventually become sufficiently good at chaining those myopic objectives together to pursue non-myopic goals once deployed.
7. Why is PPO not used for the main grid-world experiments (as discussed above)?

**Relation To Broader Scientific Literature:**

This paper falls into the general literature on reward hacking and different means of reward provision and agent supervision. While the idea of more myopic agents being inherently safer has been discussed in the past and is an assumption of other prior schemes for safer AI systems, to the best of my knowledge this is the first time it has been investigated rigorously.

**Theoretical Claims:**

I did check the correctness of the theoretical claims, and I believe they are all correct.

If I were being pedantic, I would say that the CID in Figure 2.b) should probably be such that the approval reward is downstream of some other variable (perhaps representing a common-sense understanding of the environmental dynamics or task structure) that is an ancestor of the future states or rewards, as this seems fundamental to the idea of what approval rewards are actually useful for. This clearly doesn't affect the presence of control incentives, but might matter for other analyses. (With that said, presumably if the reward function Figure 2.a is also designed deliberately then their might be similar nodes/edges there and whole thing might get a bit unwieldy, so I'm not suggesting that the authors necessarily modify the diagram, only that they consider this point.)

---

> ### Author Rebuttal · Authors · 2025-04-01
>
> Thanks for the review. We’re glad you like the paper!
>
> We’d like to clarify the role of the approval feedback in MONA and in our experiments specifically, as this partially addresses the two concerns you brought up.
>
> > I had an uneasy intuition that we were sort of just 'passing the buck' of reward specification to the approval reward, and moreover that having an approval reward capturing what the overseer views to be "reasonable" isn't all that robust. E.g. for the loan applicant summaries in Figure 6, both of these seem pretty reasonable to me, at first glance.
>
> You are right that defining a good approval reward is important in general. However in most of our experiments we focus on the effect of myopic optimization on the training incentives and do not define the approval feedback specifically for MONA (the gridworld is the only exception).
>
> In particular, in the Loan Applications experiments, the only reward on the loan summary is a monitoring penalty for explicitly using the applicant’s nationality. The reason the MONA agent avoids learning to steganographically encode nationality in the summary is because this is not explicitly reinforced and it is _not_ because the approval feedback penalizes encoding nationality (which it doesn’t).
>
> We could also define the approval feedback based on which summary an overseer views to be “reasonable” as you describe. When training with MONA, this feedback might result in the agent learning that both summaries are equally valid and it might sometimes output one or the other. But as long as the overseer's feedback is not systematically correlated with the applicant’s nationality, the MONA agent will not learn to encode nationality in the application summaries.
>
> > r_A seems to be implicitly captured by some post-hoc ground-truth evaluation that highlights the shortcomings of the reward hacking policy
>
> This is only true in the gridworld environment. In both the Test-Driven Development and the Loan Applications environment we do not distinguish between r_I and r_A explicitly, and the only difference between ORL and MONA here is myopic vs. multi-turn training.
>
>
> Also, let us respond to your specific questions below.
>
> > Please could you clarify the claim that "This is a non-trivial modification, [...]?
>
> Your interpretation is correct: we think a policy conditioned on the “You are a bad programmer” prompt might be qualitatively different from a policy “You are an expert programmer”. We agree that there is likely “a lot of overlap or transfer”. The comment in the paper was only to highlight that this difference might affect the results (even though we don’t have concrete evidence that it does).
>
> > how well do the authors expect their approach will transfer to these new methods of training more powerful foundation model agents that can plan over longer time horizons?
>
> To apply MONA to reasoning models, we'd train the model myopically to produce good reasoning steps (eg. as judged by humans). Similar to the experiments in our paper, we expect this approach to underperform ordinary RL training measured by the outcome reward. In current reasoning models the outcome reward is typically trustworthy, e.g., it is the correctness of a math problem. So we don't advocate for using MONA yet; MONA will become more relevant as we start using multi-turn RL with less trustworthy reward signals.
>
>
> > why is MONA's return approximately 0.7/2 instead of 0.7 (compared to ORL's return of 1)?
>
> The decision reward is +1 for a correct decision and -1 for an incorrect decision. So the best possible reward without using nationality is roughly 0.7 * 1 + 0.3 * (-1) = 0.4. This is described in Appendix D.2.4 but we’ll make sure to highlight it in the paper more.
>
>
> > In appendix D.1.5, whats is the "TFO" agent?
>
> Sorry, this is a typo and it should be “the _MONA_ agent”.
>
>
> > Shouldn't the ORL agent end up with a higher coordination advantage compared to MONA?
>
> It’s not clear which agent should have higher coordination advantage. In MONA the steps are rewarded independently but the second step is still trained to perform well on the distribution of application summaries the first step produces, which makes some “coordination” beneficial.
>
>
> > do the authors see their method as mainly combatting reward-hacking in training or when during deployment, or both?
>
> MONA is primarily a training method. Giving a non-myopically trained agent a myopic goal is not enough to get the benefits we describe in the paper. Using MONA during training can also help make agents easier to oversee at deployment. For example an agent that chains myopic objectives together needs to produce intermediate results which could make it easier to catch the agent pursuing undesired goals.
>
>
> > Why is PPO not used for the main grid-world experiments (as discussed above)?
>
> The results with PPO are much noisier, as you can see in Appendix F. So to see similarly clear trends we’d need to average many PPO runs for each optimization horizon.

---

> > ### Comment · Reviewer_xDwo · 2025-04-07
> >
> > I thank the authors for their response, and I apologise that my reply is quite late, as it doesn't give the authors much time to follow up. All my other questions apart from the ones below have been satisfactorily addressed.
> >
> > ## $r_A$ and $r_I$ in LLM Experiments
> >
> > > In both the Test-Driven Development and the Loan Applications environment we do not distinguish between r_I and r_A explicitly
> >
> > I _think_ I understand this now, but I still believe the paper is slightly confusing on this point. In particular, in their response I take the authors to be saying something like "actually, in the LLM experiments we don't really have $r_A$ when we train MONA, as $r_A$ is the non-myopic bit of the reward and we only train the agent myopically" (though they should definitely correct me if I am wrong), but then in the paper they also say:
> >
> > > To apply MONA, we can set $r_I = r$, but we **need to** add an additional non-myopic approval reward $r_A$ before performing myopic optimization according to (3)
> >
> > (emphasis mine) which seems to suggest that $r_A$ (i.e. the non-myopic approval reward) is **necessary** in order to be able to apply MONA. If I am not mistaken in my interpretation, then perhaps a slight re-wording of this sentence and/or a clarificatory footnote along the lines of the authors' rebuttal above would help.
> >
> >
> > ## Coordination Advantage
> >
> > > It’s not clear which agent should have higher coordination advantage. In MONA the steps are rewarded independently but the second step is still trained to perform well on the distribution of application summaries the first step produces, which makes some “coordination” beneficial.
> >
> > I still didn't get this sorry. In the paper it says:
> >
> > > Not propagating reward from the decision outcome to the application summary causes MONA to learn to write neutral summaries in Step 1 but then learn to make the correct loan decision in Step 2. So the decision making in Step 2 is crucial for achieving a high reward, which leads to a high coordination advantage.
> >
> > In particular, regarding the second sentence, does the MONA agent perform better in step 2 perform better simply because it is being trained instead of being frozen? If so I would expect basically any training scheme (that works) to lead to a coordination advantage, because a coordination advantage will (assuming having a reasonable summary to work from) boil down to to the advantage from training the agent to succeed in step 2. If instead the frozen model _has_ already been trained to perform step 2 well, where is the MONA agent's advantage coming from?

---

> > > ### Author Response · Authors · 2025-04-08
> > >
> > > Thanks for the response! Let us try to address your remaining questions below.
> > >
> > > > "actually, in the LLM experiments we don't really have $r_A$ when we train MONA, as is the non-myopic bit of the reward and we only train the agent myopically" (though they should definitely correct me if I am wrong)
> > >
> > > This is almost correct. We don't explicitly define $r_A$, but our rewards do sometimes have a non-myopic component implicitly. For example, the test-correctness in the Test Driven Development environment is not a purely myopic reward; our "test correctness" reward says something about how useful the tests are for solving the problem later so it is more like $r_A$ than $r_I$. This is what we mean in the paper, when we say "Therefore, in practice, we may choose to implement a mechanism to provide $r_{MONA}$ without explicitly distinguishing between $r_I$ and $r_A$."
> > >
> > > > (emphasis mine) which seems to suggest that (i.e. the non-myopic approval reward) is necessary in order to be able to apply MONA. If I am not mistaken in my interpretation, then perhaps a slight re-wording of this sentence and/or a clarificatory footnote along the lines of the authors' rebuttal above would help.
> > >
> > > We agree this sentence is confusing and will adapt it. Thanks for pointing this out! What we want to say here is more along the lines of "in most environments without a non-myopic approval reward myopic optimization will likely not be competitve".
> > >
> > > > In particular, regarding the second sentence, does the MONA agent perform better in step 2 perform better simply because it is being trained instead of being frozen? If so I would expect basically any training scheme (that works) to lead to a coordination advantage, because a coordination advantage will (assuming having a reasonable summary to work from) boil down to to the advantage from training the agent to succeed in step 2. If instead the frozen model has already been trained to perform step 2 well, where is the MONA agent's advantage coming from?
> > >
> > > Yes this is right. The way we set up this metric we can't really distinguish between "coordination advantage" and "learned decision making advantage". We're really missing an ablation here where we freeze step 1 and train step 2 (our frozen step 2 model is not trained at all). That's a good point and we'll clarify this in the paper!
> > >
> > > The main point of this experiment was to show that the ORL agent not just learns a single-step strategy for encoding the decision in the sentiment of the summary but that it also learns how to exploit this better in decision making. To check that the distinction between "coordination" and "learned decision making" is not as important.
> > >
> > > ---
> > >
> > > Thanks again for your constructive feedback. These are important clarifications and they will help improve the paper!

---

### Decision · Program_Chairs · 2025-05-01

**Decision:**

Accept (poster)

**Comment:**

The proposed approach, MONA, aims to prevent agents from learning long-horizon reward-hacking strategies that would be too difficult for a human supervisor to detect. The reviewers generally found the problem well-motivated and the approach well-executed.

The reviewers' remaining concerns are mostly about the experiments, which are done in small, toyish environments like Gridworld. To further address them, the metareviewer strongly encourages the authors to do the ablation referred to by reviewer xDwo in the (Final) Reply to the Authors.

However, even as is, this work definitely passes the acceptance bar.